# Randomization in clinical trials with small sample sizes using group sequential designs

**Daniel Bodden**[1]*, **Ralf-Dieter Hilgers**[1], **Franz König**[2]

**1** Institute of Medical Statistics, RWTH Aachen University, Aachen, Germany, **2** Institute of Medical Statistics, Center for Medical Data Science, Medical University of Vienna, Vienna, Austria

* daniel.bodden@rwth-aachen.de

**Data availability statement:** All code related to the simulation study is available in the following repository: Bodden D, Hilgers RD, König F. Simulation code for evaluating randomization

## Abstract

Background: Group sequential designs, which allow early stopping for efficacy or futility, may benefit from balanced sample sizes at interim and final analyses. This requirement for balance limits the choice of admissible randomization procedures. We investigate if the choice of randomization procedure, whether balanced or not, impacts the type I error probability and power in small sample clinical trials with group sequential designs.

Methods: We start with a literature review to assess how randomization procedures are reported in group sequential trials. We then investigate the impact of randomization procedures on the type I error probability and power of trials with Pocock, O'Brien-Fleming, Lan-DeMets and inverse normal combination test designs.

Results: Our findings show that only a limited number of published group sequential trials report sufficient randomization details. Simulation results demonstrate that deficiencies in the implementation of randomization can inflate type I error rates. Some combinations of group sequential designs and randomization procedures cause a loss of power, for example, when using inverse normal combination tests. When the planned balanced allocation ratio in (interim) analyses cannot be ensured, the Lan-DeMets approach is preferable for small sample trials due to its robustness to deviations between the planned and observed allocation ratio. The inverse normal combination test, while useful in trials with limited prior information, should be used cautiously with permuted block randomization that maintains the planned allocation ratio to avoid power loss.

Conclusion: We propose a framework for selecting the most suitable combinations of group sequential design and randomization procedure for small sample clinical trials. Our findings highlight the need for improved reporting of randomization methods in group sequential clinical trials. Further, the validity of some of the group sequential designs relies on the application of appropriate randomization procedures which were difficult to identify in the literature and is possibly violated.

procedures in group sequential designs; 2025. DOI: 10.6084/m9.figshare.28732184.v4.

**Funding:** RDH is coordinator and FK is member of RealiseD supported by the Innovative Health Initiative Joint Undertaking (IHI JU) under grant agreement No 101165912. The JU receives support from the European Union's Horizon Europe research and innovation programme and COCIR, EFPIA, Europa Bío, MedTech Europe, and Vaccines Europe. Views and opinions expressed are those of the author(s) only. This publication reflects the author's views. They do not necessarily reflect those of the Innovative Health Initiative Joint Undertaking and its members, who cannot be held responsible for them. RDH received funding from European Rare Disease Research Coordination and Support Action consortium (ERICA) funded by the European Union's Horizon 2020 research and innovation programme under Grant Agreement. no. 964908. The funders had no role in study design, data collection and analysis, decision to publish, or preparation of the manuscript.

**Competing interests:** The authors have declared that no competing interests exist.

## Introduction

In traditional clinical trial design, patient enrollment continues until a predetermined sample size is reached. Group sequential designs (GSDs) offer an alternative by allowing for interim analyses at predetermined points, for example by time or achieved sample size during the trial. These interim analyses provide the possibility of stopping the trial early, either for efficacy if the treatment shows a significant benefit, or for futility if continuing the trial is unlikely to yield conclusive results.

While traditional GSDs are commonly applied in larger confirmatory trials, they are increasingly being suggested for small population clinical trials, as noted by EMA's Committee for Medicinal Products for Human Use (CHMP) guideline [1] and other published research in the field of rare diseases [2–5]. Sequential designs can be particularly advantageous in these settings by enabling early stopping for efficacy or futility, which is especially valuable when patient recruitment is challenging or when ethical considerations, such as those in orphan medicine, demand minimizing participant exposure. In some cases, treatments for rare diseases can demonstrate such pronounced efficacy that the required sample size may be considerably lower than initially anticipated [6]. In these situations, GSDs can provide an essential mechanism for early trial termination during interim analyses [4]. A systematic review of randomized clinical trials in the field of cardiovascular clinical trials [7] found that 9.3% (4 out of 43) of clinical trials with a traditional GSD had a sample size smaller than 200.

However, interim analyses introduce a multiple testing problem, so control of the overall significance level $\alpha$ is a critical aspect [8], addressed by various procedures [9,10].

Prominent GSDs proposed by Pocock [11] and O'Brien and Fleming (OBF) [12] adjust the significance level for each stage to control the overall significance level $\alpha$. Pocock's design distributes adjusted significance levels equally across stages, whereas OBF's design starts with a lower significance level at early stages, gradually increasing it with each successive test.

In their original formulations, Pocock and OBF designs assume a fixed number of stages and equidistant timings usually associated with the fraction of enrolled patients so far for interim analyses. These designs usually assume balanced treatment allocations at each stage [11,12]. The impact of violating these assumptions has been investigated [9,11,13].

Lan and DeMets (LDM) proposed a more flexible GSD by introducing an alpha spending function based on the observed information fraction for each stage, allowing for adjustments in the timing and number of interim analyses [14]. This alpha spending function can be chosen to closely resemble the boundaries of both the OBF and Pocock designs, often referred to as "OBF spending function" or "Pocock spending function."

The inverse normal combination test (INCT) ensures control of the overall type I error rate (T1E) even when the sample size is adjusted based on unblinded interim treatment effect estimates, by combining p-values from different stages [15]. An advantage is that group sequential boundaries, for example coming from LDM spending functions, can be directly applied.

Regarding the application of GSDs, a systematic review [16] of randomized controlled trials published between 2001 and 2014 found that only 65% of studies reported the stopping rules or boundaries used. Among these, 34% used the LDM design with an OBF error spending function, while 27% used the traditional OBF design. Other reported boundaries included those proposed by Pocock [11], Haybittle Peto [17], Pampallona and Tsiatis [18]. The exact percentages for the use of these designs were not specified in the systematic review [16].

Randomization is a fundamental component of clinical trial design, serving to eliminate biases, ensure comparability of treatment groups and validity of the test decision [8,19,20]. In contrast to often poorly documented randomization procedures (RPs) in general clinical trials

[21], 84% of GSD trials provided "complete reporting" of the type of randomization, although details are not given [16].

Obviously, without further restriction on the randomization process, the number of patients allocated to the treatment groups at the respective interim analysis points or at the end of the trial may vary, which may influence the T1E and power of a GSD. Thus, it is important to investigate whether the interaction between RPs and different GSD rules impact the performance of the trial.

The objective of this paper is to evaluate how different RPs affect T1E and power in small sample group sequential clinical trials. Specifically, we investigate both balanced and unbalanced RPs (during interim and final analyses) and evaluate how they affect Pocock, OBF, LDM and INCT designs. The focus on small population trials is important, as imbalances in randomization can have a more pronounced impact in these settings.

The structure of this paper is as follows: The "Methods" section provides the setup for a literature review on randomization in group sequential trials and outlines the methodology of the simulation study conducted. The "Results" section presents the findings from both the literature review and the simulation study, offering a framework for selecting the appropriate combination of GSD and RP in small sample clinical trials. The "Discussion" section examines the advantages and disadvantages of various RPs and GSDs, and explores their implications for future research. Finally, the "Conclusion" section summarizes the key findings on the use of randomization in GSDs.

## Methods

We structure the Methods section as follows: First, we describe the literature review setup. This is followed by a description of the clinical trial design and RPs. Next, we outline the adjustment for repeated interim analyses followed by how a t-test can be performed for a GSD. We conclude the section with the simulation study setup used to evaluate T1E and power. The results corresponding to each of these subsections are presented separately in the Results section.

### Literature review setup

To establish the relevance of this paper, we conducted a literature review to explore the aspects of randomization and balancing in GSDs, aiming to build upon the findings of Stevely et al. [16], who assessed adherence to the CONSORT guideline in GSDs.

Identifying studies implementing GSDs proved challenging due to inconsistent reporting standards. Although the recent CONSORT extension for adaptive designs recommends including the term "adaptive" in the abstract or as a keyword [22], this practice has not been widely implemented yet. Therefore, we applied a similar search strategy to Stevely et al. [16]. We conducted the search on the Ovid MEDLINE database, covering the period from January 1, 2019, to May 19, 2023. Our keywords were "group sequential," "interim analysis," and "interim analyses," as identified in [16]. We combined these keywords using the Boolean operator "OR" and included the following additional eligibility criteria: human subjects, full-text, and English language. For the publication type filter, we used "Clinical trial, all", in contrast to Stevely et. al. [16], who restricted their selection to "Clinical trial, Phase III". Our broader filter was chosen to include randomized Phase II clinical trials, as these are particularly relevant for rare disease clinical trials [23]. To determine the RP used in each study, we also considered supplementary materials, namely the statistical analysis plan and study protocol when accessible.

## Clinical trial design and randomization procedures

We consider a randomized controlled single center trial with normal endpoint using a group sequential two arm parallel group design with an intended allocation ratio of $1:1$. The interim analyses are assumed to be equidistantly spaced. We are interested in testing the following one-sided null hypothesis $H_0 : \mu_E \leq \mu_C$ against the alternative hypothesis $H_1 : \mu_E > \mu_C$ using a one-sided z-test at level $\alpha$, assuming known common variance of 1, where $\mu_E$ and $\mu_C$ denote the expected response of the experimental and control group.

The impact of randomization-induced imbalances in treatment allocations within GSDs has not been investigated, since the actual allocation ratio at interim analyses depends on the implemented RP. Some of the RPs can be specified to ensure balanced treatment allocations at interim time points, while others need similar restrictions like in stratified randomization [24]. Table 1 provides an overview of key terms related to randomization in GSDs. Specifically, the RPs evaluated in this study are defined, providing a reference for the methods under consideration. The RPs will be applied across the entire trial rather than being implemented separately for each stage.

## Adjustment for repeated interim analyses

Table 2 shows important terms related to GSDs. Traditional GSDs have been introduced for equally spaced interim analyses, so that the number of patients between two stages is the same [11,12]. However, with LDM spending functions in GSDs flexible timing of interim analyses was allowed while controlling the T1E through a pre-defined spending function. This dictates

**Table 1. Glossary of important terms related to randomization in 2-arm clinical trials comparing two treatments.**

| Term | Description |
|---|---|
| **Terms related to randomization** | |
| Randomization | A method which introduces a random element when assigning a treatment to a patient in a randomized controlled clinical trial. |
| Randomization sequence | The order in which participants are assigned to treatment groups in a randomized clinical trial. The sequence is typically generated from a randomization procedure to ensure unbiased treatment allocation [8]. |
| Allocation ratio | The allocation ratio specifies the proportion of participants assigned to each group in a trial at the time of enrollment. The actual allocation ratio achieved by the randomization method may differ from the planned or intended ratio of the study, which is usually defined for the end of the trial. If the allocation ratio equals 1 the allocation is named *balanced*. |
| Complete Randomization (CR) | Randomization achieved by flipping a fair coin. Sometimes this method is also referred to as simple or full randomization [25,26]. |
| Random Allocation Rule (RAR) | Randomization assigning the same proportion of patients to each treatment [25,27]. Note that the abbreviation "RAR" refers to the random allocation rule in this paper and should not be confused with response-adaptive randomization. |
| Permuted Block Randomization (PBR($l$)) | Allocation in blocks of length $l$, with randomization within each block according to RAR. See, for example [25,28]. |
| Randomized Permuted Block Randomization | Permuted block randomization with block sizes for each block randomly selected from a predefined set [25]. |
| Efron's Biased Coin (EBC($p$)) | Randomization using a biased coin with probability $p$ in favor of the treatment with fewer allocations and a fair coin in case of equal allocations [25,29]. |
| Big Stick Design (BSD($m$)) | Complete randomization with deterministic assignment when a maximum tolerated imbalance $m$ is reached [25,30]. |
| Chen's Design (Chen($p,m$)) | EBC($p$) with deterministic assignment when a maximum tolerated imbalance $m$ is reached [25,31]. |

**Table 2. Glossary of important terms related to group sequential designs.**

| Term | Description |
|---|---|
| **Terms related to interim analyses and group sequential designs** | |
| Interim Analysis | "Any analysis intended to compare treatment arms with respect to efficacy or safety at any time prior to the formal completion of a trial.", cited from [8]. |
| Group Sequential Design (GSD) | A randomized clinical trial with pre-planned interim analyses and the option to stop the trial early for efficacy and/or futility based on pre-defined adjusted boundaries to control overall the type I error rate at the nominal level $\alpha$, see, for example, [9,10]. |
| Efficacy Boundary | An efficacy boundary in GSD is a threshold to stop the trial early for efficacy usually accounting for the repeated significance testing at interim and final analyses. |
| Futility Boundary | A futility boundary in GSD is a threshold to stop the trial early for lack of efficacy. Two types are distinguished: binding and non-binding [33]. With binding futility boundaries, the trial must be stopped if these boundaries are crossed, which allows for lower efficacy boundaries. In contrast, non-binding futility boundaries allow flexibility, although they suggest stopping for futility, the decision can be overruled. However, when using non-binding rules, the potential of early stopping for futility is not factored into the calculation of efficacy boundaries. |
| Pocock | A type of efficacy boundaries [11] in a GSD that adjusts the significance level equally for each stage to control the overall significance level $\alpha$. Originally proposed for GSD with equally sized stage-wise sample sizes resulting in the same boundaries across all stages. |
| O'Brien-Fleming (OBF) | A type of efficacy boundaries [12] in a GSD that adjusts the significance level for each stage to control the overall significance level $\alpha$. The design starts with a low significance level at early stages, gradually increasing it with each successive test. |
| Lan-DeMets (LDM) | A more flexible way to derive boundaries in a GSD using a pre-defined alpha spending function [14] based on the observed information for each stage. This allows for unequal stage-wise sample sizes and adjustments in the timing and number of analyses. The alpha spending function can be chosen to resemble OBF and Pocock type boundaries. |
| Inverse Normal Combination Test (INCT) | The INCT [15] combines stage-wise test statistics using a predefined combination function, rather than accumulating data as in standard GSD. INCT has the advantage of allowing standard GSD boundaries, as described above, while also permitting data-dependent adaptations [10,32,33], such as adjustments to the allocation ratio and sample size, all while strictly controlling the type I error rate. |

how much error rate is "spent" at each interim stage, for example, by approximating Pocock or OBF type of boundaries [14]. These GSDs are commonly used to provide a framework to stop early for efficacy or futility based on cumulative data. The main idea is to use all accumulated data collected so far for the calculation of a standard test statistics at each (interim) analysis and adjust the critical boundaries accordingly for repeated significance testing. In contrast, GSDs using the inverse normal combination test (INCT) combine stage-wise p-values or other test statistics across multiple stages. To control the T1E, the stage-wise information is combined using pre-defined weights. An advantage of using the INCT is, that the same critical boundaries as for standard GSDs can be used [15,32].

The INCT enables a more flexible design [32], such as, adjusting sample sizes for the next stages based on the data observed so far. For instance, these adjustments can be made based on conditional power arguments [34]. For the study protocols the critical boundaries are usually determined for the anticipated number of available subjects at each interim analyses and a fixed allocation ratio, for example, $1:1$ allocation. In this paper $n$ denotes the maximum sample size and $K$ the number of stages for the GSD (i.e. the number of analyses conducted), for example, $K = 3$ corresponds to a trial with two interim analyses and a final analysis.

Fig 1 illustrates test statistics and boundaries for a GSD with a maximum sample size of $n = 24$ and $K = 3$ stages. The figure compares a planned $1:1$ allocation ratio at each stage with an observed allocation that deviates from this intended ratio. In Fig 1, deviations from the $1:1$ allocation ratio in stages 1 and 2 result in reduced information.

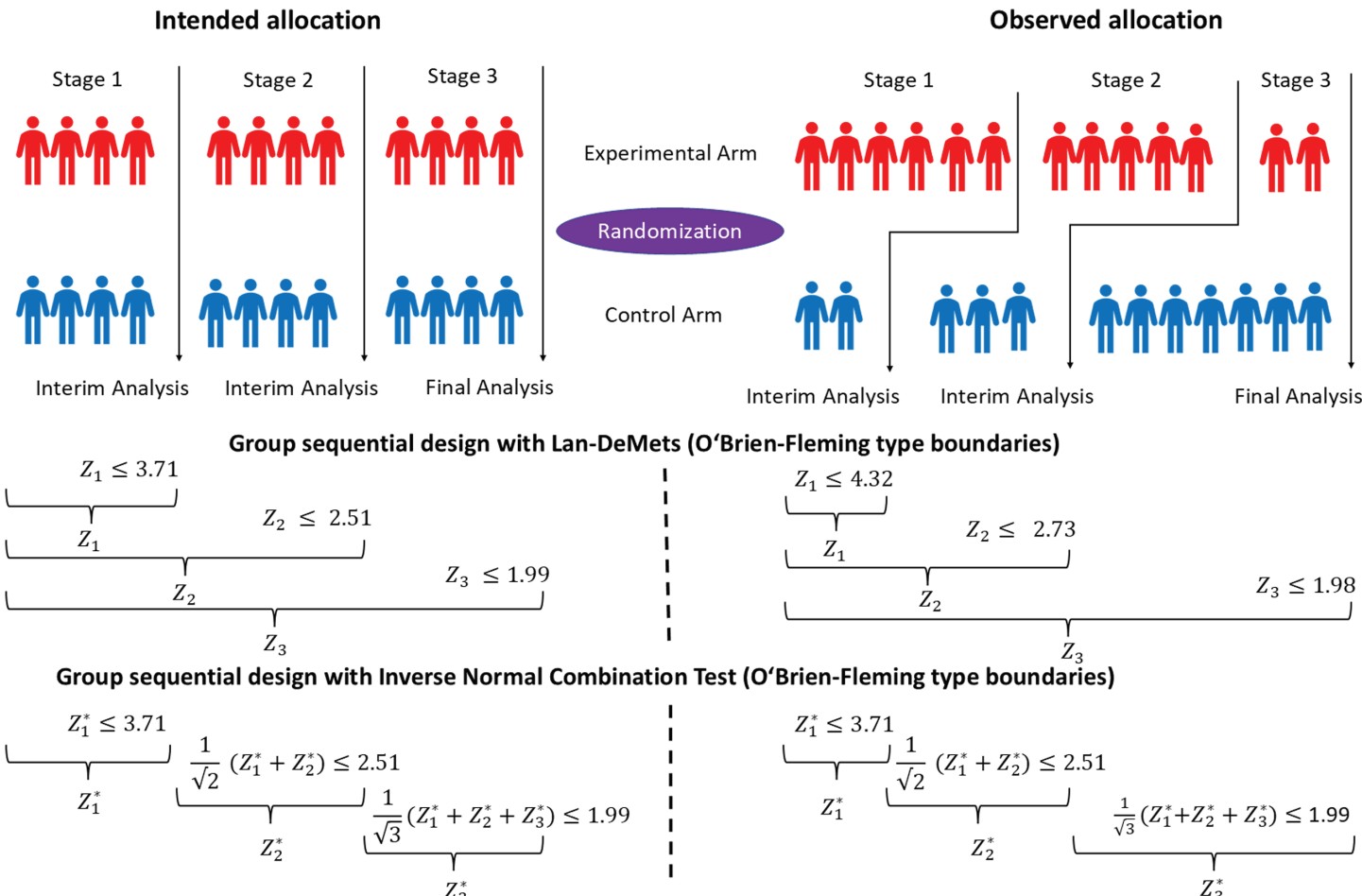

**Fig 1. Differences between planned allocation and observed allocation in group sequential designs.** Comparison of Lan-DeMets and the inverse normal combination test, both using O'Brien-Fleming type boundaries. The figure illustrates the differences between the planned allocation, which strictly follows a 1 : 1 ratio at each interim and final analysis, and the observed allocation, where stage 1 allocates 6 patients to the experimental arm and 2 to the control arm, stage 2 allocates 5 to the experimental arm and 3 to the control arm, and stage 3 allocates 2 to the experimental arm and 6 to the control arm. $Z_i$ denotes the cumulative test statistic of all patients up to stage $i$ used in Lan-DeMets designs and $Z_i^*$ denotes the stage-wise test statistic used in the inverse normal combination test.

In the LDM approach, the information accumulated up to each stage is used to determine the boundaries. However, if the observed allocation deviates from the intended 1 : 1 ratio, the actual information collected will be less than expected and the boundaries need to be adjusted.

While both methods ensure strong control of T1E, if this assumption holds, only the INCT controls the T1E in case of deviations, when sticking to the initially calculated boundaries. If for traditional GSDs the initially calculated boundaries are used, but the observed allocation ratio differs from the initially planned allocation ratio, this can elevate the T1E. Therefore, this has to be addressed in the spending function.

Thus we will calculate the T1E and power for different GSDs and randomization sequences from various RPs. We will consider the mean (overall) power as well as mean T1E across simulated sequences of a RP. A technical description of how the T1E and the power, conditioned on a randomization sequence, were calculated using the multivariate normal function

is provided in S1 Appendix and S2 Appendix for both traditional GSDs and GSDs with the INCT.

The appendix also details the incorporation of futility boundaries, which can be used to stop the trial early when a significant result is unlikely. Futility boundaries can be defined as either binding or non-binding. Binding futility boundaries require that the trial is stopped if crossed, while non-binding boundaries provide the option to stop the trial. If binding futility boundaries are used, the (interim) efficacy boundaries also need to be adjusted. If non-binding futility boundaries are used, then the futility boundaries will be the same as for the case when no futility stopping is used, but for the calculation of the power or sample sizes the non-binding futility boundaries are used.

## Group sequential t-test

Given the small sample sizes, the t-test is generally the more appropriate choice for analysis. However, applying a t-test within a GSD is not straightforward and introduces additional challenges. To address this, we used the quantile substitution method [11]. This method transforms the z-statistic boundaries into p-values and then into t-statistic boundaries to compare them with the t-statistics calculated for the respective stages. While this method offers a practical solution, it is important to note that it can lead to an increased T1E [35]. Details on the methodology for the group sequential t-test are provided in S8 Appendix. Alternative methods, such as determining boundaries via Monte Carlo simulation [36] or through exact calculations [9,37], are available but are rarely used due to their high computational demands.

## Simulation study setup

In our simulation study, we generated randomization sequences based on a given RP and evaluated the T1E conditioned on these sequences for various GSDs. Similarly, we assessed the power conditioned on the randomization sequence for different effect sizes and GSDs. To calculate the power of a GSD using a specific RP, we used the mean power conditioned on the generated randomization sequences. It is important to note, that when we refer to "power of a GSD" in this paper, we are referring to the mean power conditioned on the randomization sequences generated for a given RP.

For the OBF, Pocock and LDM designs, we excluded randomization sequences where patients were only allocated to a single treatment group in the first stage, as in that case the "information" cannot be calculated, as detailed in S1 Appendix. Similarly, for INCT, we excluded randomization sequences, where any stage included patient allocations to a single treatment group only, as explained in S2 Appendix. The frequency of these exclusions was tracked and can be found in the S3 Appendix.

We simulated 100,000 randomization sequences for each RP and GSD for the T1E. For assessing the power, we generated 1000 randomization sequences for each RP and GSD across a grid of effect sizes ranging from 0 to 2 in increments of 0.2. This number of randomization sequences was found to be sufficient, as shown by the maximum standard error, which is given in the respective tables.

The following parameters were used for the RPs: For Efron's Biased Coin (EBC), we chose the parameter $p = \frac{2}{3}$. For Big Stick Design (BSD), we selected a maximum tolerated imbalance $m = 3$, as recommended in [38]. Similarly, for Chen's design, we used $p = \frac{2}{3}$ and $m = 3$. As for Permuted Block Randomization (PBR), we selected a block size of $l = 4$, the most commonly used block size, as identified in our literature review. Please note that for the simulation study all RPs are implemented for the maximum sample size, for example, if block randomization with larger block length than the stage-wise sample size was used, a block can

overlap over two stages. Similar if restricted methods were used, the restriction applies to the maximum size, but not to the stage. We also evaluated GSD with binding and non-binding futility boundaries as the FDA recommends the use of non-binding futility boundaries when determining efficacy boundaries in group sequential and adaptive designs [39].

S3 Appendix provides a Table with an overview of all simulations performed, including their respective parameters and settings for the z-test.

Following reviewers suggestion, we simulated the T1E and power when using a t-test. To achieve this, we first generated 1000 randomization sequences for each RP. Next, we simulated 2000 clinical trials and calculated the mean T1E for each randomization sequence within each RP. For LDM designs, randomization sequences, which allocated less than two patients to treatment or control group in stage 1 were skipped, as for these sequences the t-test cannot be conduced. For INCT designs, all randomization sequences that allocated less than two patients to any group and any stage were skipped, as the t-test is calculated separately for each stage for INCT. The number of excluded sequences of our simulation is provided in S8 Appendix.

To generate randomization sequences from various RPs, we used the R package randomizeR [40]. For group sequential boundaries, we employed the gsDesign package [41] and the rpact package [42], The calculation of the multivariate integral needed for the simulation was performed using the mvtnorm package [43]. The simulations were executed using R version 4.2.2 on the National High Performance Computing Center for Computational Engineering Science [44]. The code related to the simulation study is available on figshare [45]. Validation of the implementation can be seen by comparing the results obtained using the multivariate normal integral with those from the clinical trial simulations for the z-test, as detailed in S9 Appendix.

## Results

### Results of the literature review

The process of study identification is illustrated in Fig 2. In total 71 studies were included in our review with the summarized results presented in Table 3. An Excel file containing all included studies and their respective settings is available in S11 Appendix.

52% of the studies employed an LDM approach with an OBF type alpha spending function. The OBF design was used in 13% of studies. Thus overall OBF type of boundaries were the most commonly used ones. This is not an unexpected finding, as OBF GSD require almost no increase in the maximum sample size compared to designs with a fixed sample size as most of the $\alpha$ is spent in the final analysis anyhow. Designs with Pocock type boundaries were less common, making up 1% of studies for the LDM approach with Pocock type boundaries and 3% for the Pocock design. The GSDs categorized under "other" can be found in S11 Appendix

In terms of RPs, the most utilized method was PBR, found in 75% of the studies that reported their RP. Minimization was used in 23% of the studies reporting the RP, and a single study reported using "simple randomization." The most frequent block size for PBR was 4. Stratification was implemented in 76% of the studies. For our review, we considered minimization a form of stratification. A 1 : 1 allocation ratio was used mostly, found in 83% of studies. This can be explained by the fact, that when deviating from a 1 : 1 allocation ratio, usually higher sample sizes would be required to achieve the same power. Most studies had a planned sample size of at least 1000, with only three trials having fewer than 101 planned participants. This highlights, despite being promoted in regulatory guidance's and methodological papers for rare-diseases, that GSDs are not widely adopted in trials with (very) small sample sizes.

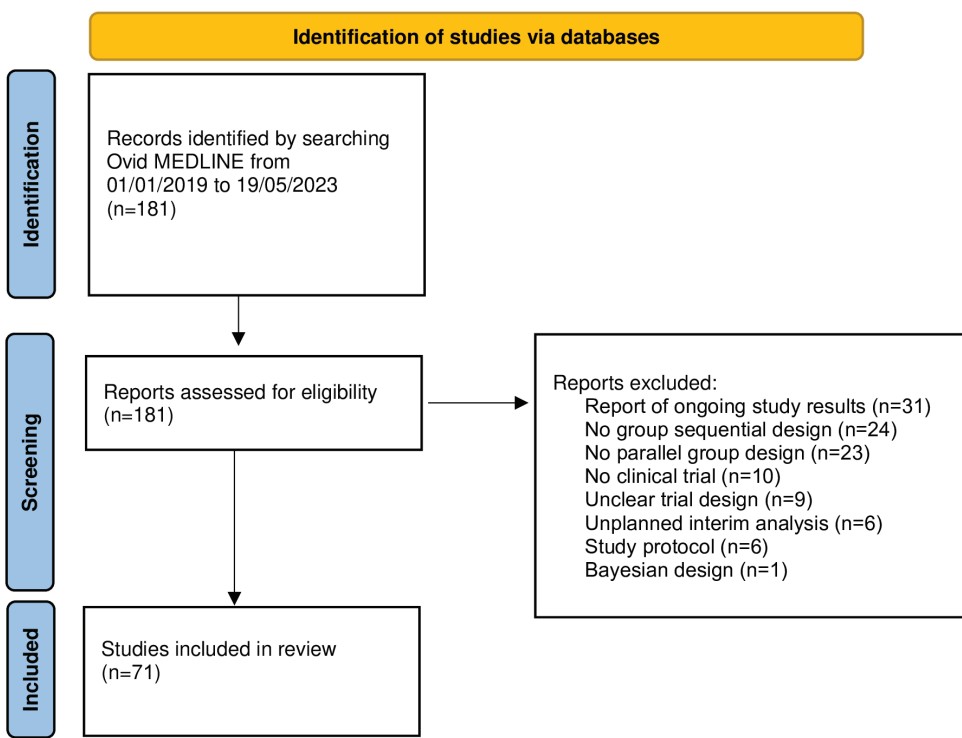

**Fig 2. Adjusted PRISMA 2020 Flowchart for the conducted literature review.** The review explored randomization and balancing aspects of group sequential designs. The search was conducted in Ovid MEDLINE from January 1, 2019, to May 19, 2023, using the input "group sequential OR interim analysis OR interim analyses" and applying additional eligibility criteria: human subjects, full-text, and English language.

**Results of the simulation study for the z-test.** Consider the following scenario: In a trial with a GSD, the critical boundaries are specified in the study protocol based on the (planned) stage-wise sample size and number of stages. These predefined boundaries are then naively applied during the analysis, without accounting for the stage-wise deviations in the observed allocation ratio due to the RP. Fig 3 illustrates this scenario with a violin plot depicting the T1E for a maximum sample size of $n = 24$ and 2 interim analyses for different RPs. This increase in T1E occurs when the intended allocation ratio is not maintained for every stage of the trial.

The largest inflation in mean T1E is observed for RAR, as balanced allocation at the interim analysis is not always achieved, even though RAR results in an overall 1 : 1 allocation by the end of the trial.

All evaluated RPs led to a mean T1E increase except for PBR(4), as PBR(4) ensures a 1 : 1 allocation ratio for each stage. Consequently standard Pocock and OBF stopping boundaries should only be used in conjunction with PBR when the block size is a divisor of the stage-wise sample sizes to ensure the 1 : 1 allocation ratio is preserved. In the case of randomized PBR, all block sizes should likewise be divisors of the stage-wise sample sizes to maintain balanced allocation ratios for each stage. However, even when a stage-wise balanced RP is employed, practical issues such as missing data or participant dropout may disrupt the intended allocation ratio.

**Table 3. Summary of results from the literature review.**

| Parameter | Total (%) |
|---|---|
| **Group sequential design and upper boundary** | n = 71 |
| Lan-DeMets with O'Brien-Fleming alpha spending† | 37 (52) |
| Lan-DeMets with Pocock alpha spending | 1 (1) |
| O'Brien-Fleming design | 9 (13) |
| Pocock design | 2 (3) |
| Inverse normal combination test with O'Brien-Fleming | 1 (1) |
| Boundaries given only | 10 (14) |
| Other | 9 (13) |
| No information given | 2 (3) |
| **Randomization Procedure** | n = 71 |
| (Randomized) Permuted Block Randomization | 30 (42) |
| No information on block length | 16 |
| Information on block length(s) | 14 |
| 4 | 6 |
| 10 | 2 |
| 20 | 1 |
| 3, 6 | 1 |
| 4, 6 | 1 |
| 2, 4, 6 | 1 |
| 2, 4, 6, 8 | 1 |
| 4, 6, 8, 10 | 1 |
| Minimization | 9 (13) |
| Simple Randomization | 1 (1) |
| No information given | 31 (44) |
| **Allocation Ratio** | n = 71 |
| 1:1 | 59 (83) |
| 2:1 | 11 (15) |
| 3:2 | 1 (1) |
| **Stratification*** | n = 71 |
| Yes | 54 (76) |
| No | 17 (24) |
| **Planned sample size** | n = 71 |
| 1–100 | 3 (4) |
| 101–200 | 11 (15) |
| 201–500 | 19 (27) |
| 501–1000 | 17 (24) |
| ≥1001 | 20 (28) |
| No information | 1 (1) |

†One oncology study had two primary endpoints overall survival (OS) and progression free
survial (PFS), each tested in the overall study population and in a pre-defined subgroup. Here
we included the boundary for the more important endpoint OS, which used Lan-DeMets
with O'Brien-Fleming alpha spending. For PFS a different spending function was used.
*Minimization is considered a form of stratification here.

Both LDM and INCT maintain exact control over the T1E, even when deviations from
the 1 : 1 allocation ratio occur at different stages. This was confirmed through simulations
presented in S4 Appendix.

Another issue identified in the simulation is the potential for certain RPs to allocate all
patients to a single group within a stage. For LDM it is important that each group receives at
least one allocation in the first stage to enable testing across all stages. For INCT every stage
must allocate at least one patient to each group. This can be explained by the difference in

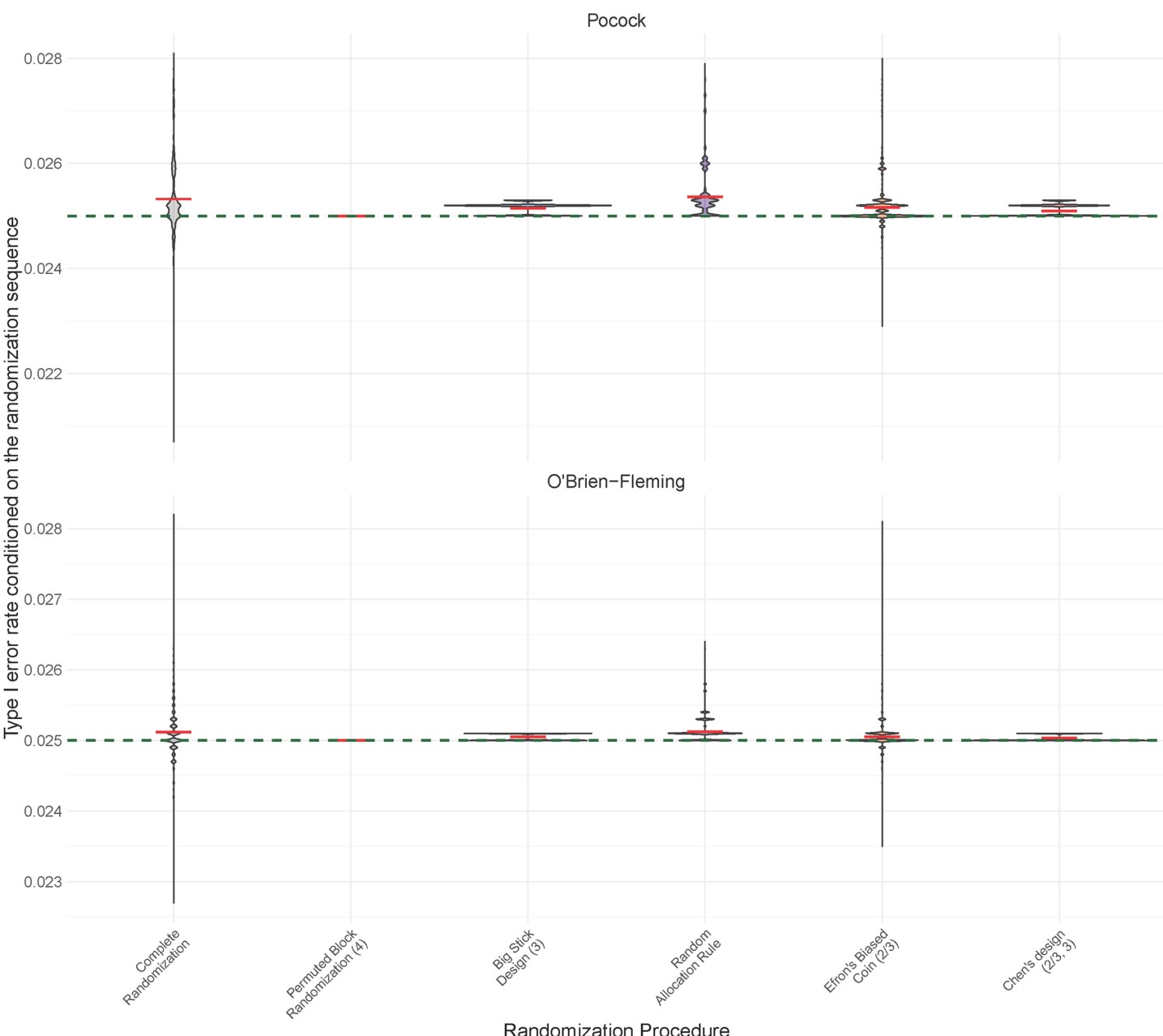

**Fig 3. Violin plots showing type I error rates conditioned on the simulated randomization sequences of different randomization procedures for the z-test.** For a maximum sample size of $n = 24$ and three equally sized stages ($K = 3$), i.e. two interim analyses and a final analysis, using standard O'Brien-Fleming (second row) and Pocock boundaries (first row). The standard O'Brien-Fleming and Pocock boundaries assume a 1 : 1 allocation at each stage. The mean type I error rate conditioned on the randomization sequence for each randomization procedure is shown by the red line. To make the inflation more visible, we zoom into the y-axis showing just the interval [0.023,0.028]. The same figure with y-axis starting with 0 is shown in the S4 Appendix not to exaggerate the findings. Additionally, the S4 Appendix has the same figures for the correct spending function acknowledging the observed allocation ratio and for the inverse normal combination test.

how test statistics are computed: LDM uses data from all patients up to a given stage, while INCT uses only patients within each stage, as illustrated in Fig 1. A more detailed theoretical explanation is provided in S1 Appendix and S2 Appendix.

For a maximum sample size of $n$ = 24 and $K$ = 3 stages, the first stage shows allocation to only one group with probability 0.0078125 for CR, with probability 0.0005 for EBC( 2/3) and with probability 0.0004 for RAR. Further details, including the number of randomization sequences skipped due to single-group allocations in the first stage for LDM or in any stage for INCT, are provided in S3 Appendix.

Additionally, when using RPs that allow a maximum tolerated imbalance $a$, this maximum tolerated imbalance should be smaller than the smallest stage size for LDM and less than half the smallest stage size for INCT. This requirement is necessary, because once an imbalance of $a$ is reached in one direction, the allocation rule allows up to $2a$ consecutive allocations to the opposite group.

We next examined how these RPs influence the power of the trial under different group sequential settings. Table 4 shows the power in a setting with $n$ = 24 and $K$ = 3 for a standardized effect size of $\delta$ = 1.0 and $\delta$ = 1.4 across different combinations of RPs and GSDs. The effect size $\delta$ = 1.0 corresponds to approximately 69% power for an one-sided z-test (without interim analyses) at a significance level of $\alpha$ = 0.025. Similarly, an effect size of $\delta$ = 1.4 corresponds to approximately 93% power under the same conditions. The power results for LDM OBF and LDM Pocock are generally similar in regards to the RP, with the exception of CR, which should be avoided both due the power loss and the potential for allocating patients to a single treatment group only in the first stage.

**Table 4. Power for standardized effect sizes of $\delta$ = 1.0 (top) and $\delta$ = 1.4 (bottom) for each combination of randomization procedure and group sequential design for the z-test.**

| Randomization Procedure | Lan-DeMets with O'Brien-Fleming type boundaries | Inverse Normal Combination Test with O'Brien-Fleming type boundaries | Lan-DeMets with Pocock type boundaries | Inverse Normal Combination Test with Pocock type boundaries |
|---|---|---|---|---|
| Power for standardized effect size of $\delta$ = 1.0 | | | | |
| **Complete Randomization** | 0.6629 | 0.6227 | 0.5916 | 0.5511 |
| **Permuted Block Randomization** (4) | 0.6819 | 0.6819 | 0.6100 | 0.6100 |
| **Big Stick Design** (3) | 0.6800 | 0.6577 | 0.6081 | 0.5859 |
| **Random Allocation Rule** | 0.6823 | 0.6396 | 0.6109 | 0.5676 |
| **Efron's Biased Coin** (2/3) | 0.6790 | 0.6517 | 0.6072 | 0.5807 |
| **Chen's Design** (2/3, 3) | 0.6807 | 0.6643 | 0.6088 | 0.5927 |
| Power for standardized effect size of $\delta$ = 1.4 | | | | |
| **Complete Randomization** | 0.9145 | 0.887 | 0.8421 | 0.8761 |
| **Permuted Block Randomization** (4) | 0.9264 | 0.9264 | 0.8906 | 0.8906 |
| **Big Stick Design** (3) | 0.9254 | 0.9121 | 0.8724 | 0.8893 |
| **Random Allocation Rule** | 0.9266 | 0.8988 | 0.8562 | 0.8912 |
| **Efron's Biased Coin** (2/3) | 0.9248 | 0.9074 | 0.8674 | 0.8886 |
| **Chen's Design** (2/3, 3) | 0.9258 | 0.9161 | 0.8776 | 0.8897 |

**Table Description:** Based on 1000 randomization sequences generated from each randomization procedure to calculate the (mean) power. For a maximum sample size of $n$ = 24 and three equally sized stages ($K$ = 3), i.e. two interim analyses and a final analysis. The maximum standard error is given by 0.0015 for inverse normal combination test with O'Brien Fleming type boundaries and complete randomization. For a description of the randomization procedures see Table 1.

For the INCT the variability in power is higher. This can be attributed to the fact that this test applies the same weights across stages, regardless of the observed allocation ratio. Notably, the power of PBR(4) is approximately 2.4 percentage points higher than that of BSD(3) for both INCT OBF and INCT Pocock and approximately 1.7 percentage points higher than that of CHEN(2/3, 3) for both INCT OBF and INCT Pocock.

PBR(4) shows equal power when using either LDM or INCT. However, the power of INCT declines when RPs that may cause stage-wise imbalances are employed. Fig 4 illustrates the power as a function of the effect size for various RPs under INCT OBF. The highest power is observed for PBR(4), followed by CHEN(2/3, 3), BSD(3), EBC(2/3), RAR and finally CR. This aligns with how strictly each RP enforces balance across the stages, with more restrictive procedures maintaining higher power. The degree of restrictiveness is detailed in S10 Appendix, which presents the absolute mean differences in group sizes between the interim and final analyses for each RP.

Next, we examined the power differences when futility stopping is used. We used a futility stopping boundary, where we stopped in case of a negative trend, i.e. the observed interim estimate of the treatment group is worse than of the control group. Table 5 shows the power for both binding and non-binding futility boundaries for $n = 24$ and $K = 3$ with a standardized effect size of $\delta = 1.0$ and $\delta = 1.4$. We included the power values for scenarios without futility boundaries for comparison.

As expected, when a binding futility boundary is applied, the power decreases slightly. The power decreases even more when a non-binding futility boundary is applied. For LDM OBF, we observed that the power remains robust for both types of futility boundaries across different RPs, with the exception of CR, which leads to a power loss in comparison to the other RPs. The differences in power for INCT OBF across the different RPs when futility boundaries are used is comparable to the case where no futility boundaries are used. The power loss for INCT OBF associated with adding a binding futility boundary is about as high as using CHEN(2/3, 3) instead of a procedure which maintains the 1 : 1 allocation exactly at every stage.

To get a deeper insight into how power is affected by different RPs across varying sample sizes, we evaluated the power as a function of the maximum sample size. Fig 5 illustrates the power for a standardized effect size of $\delta = 1.0$ for $K = 3$ equidistant stages with varying maximum sample sizes. The power was computed on a grid of sample sizes ranging from 12 to 60, with increments of 6, to ensure equal allocations between groups for each stage when $K = 3$.

The overlapping lines with the highest power are the different RPs For LDM OBF with the exception of CR. The power for LDM OBF remains similar across all evaluated RPs with the exception of CR, consistent with the results observed for $n = 24$. It is noticeable, that when varying the sample size, PBR(4) does not preserve the 1 : 1 allocation for each stage anymore.

For INCT the power shows a higher variability depending on the RP. PBR(4) still has the highest power under the RPs using an INCT design. However, the power is lower than for the LDM designs whenever $n/3$ is not divisible by 4, as for these sample sizes PBR(4) does not maintain the 1 : 1 allocation ratio for each stage. When $n/3$ is divisible by 6, we can see that the power of PBR(4) coincides between INCT and LDM, which can be seen for the maximum sample sizes $n = 12$ $n = 24$, $n = 36$, $n = 48$ and $n = 60$ in the figure.

The power differences between the RPs becomes smaller as the stage-wise sample size increases. Additional results for different maximum sample sizes and number of stages are provided in S6 Appendix. Notably, these differences become negligible when the number of patients per stage exceeds 40, with only CR and RAR still showing a notable drop in power.

PBR guarantees a 1 : 1 allocation at each stage only when the stage size is divisible by the block size. Otherwise we have to deal with a power loss for INCT as we are using equal

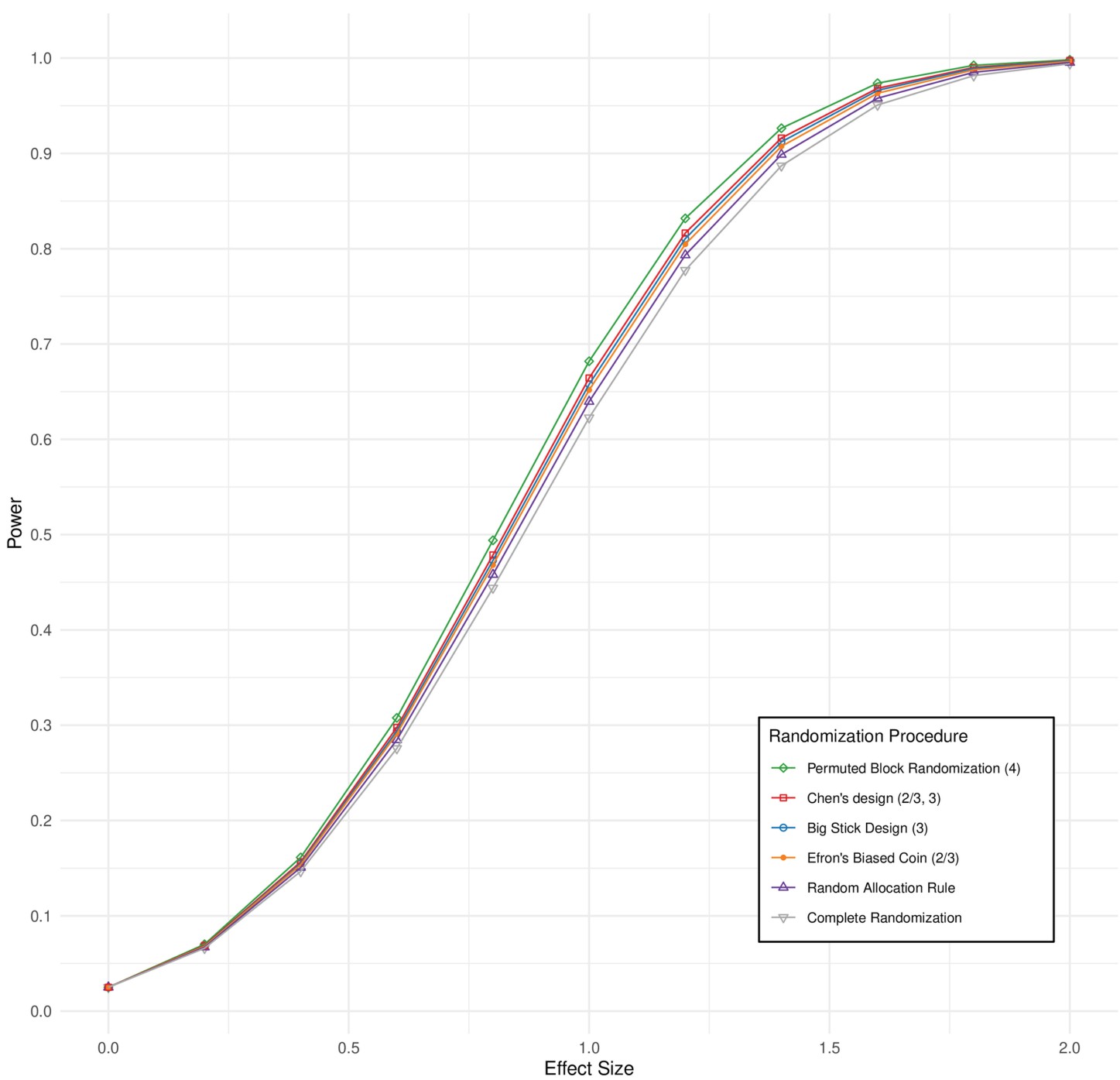

**Fig 4. Power depending on the standardized effect size for a group sequential design comparing two treatment arms using an inverse normal combination test at a nominal one-sided level of $\alpha = 0.025$ and O'Brien-Fleming efficacy boundaries for different randomization procedures for the z-test.** For a maximum sample size of $n = 24$ and three equally sized stages ($K = 3$), i.e two interim analyses and a final analyses. For the inverse normal combination test equal weights for all stages were used.

weights for all stages. However, this assumption is not possible for all sample sizes. Additionally, even small over- or under-running within stages will disrupt the 1 : 1 allocation. Therefore caution is required to not over- or under-run within the stages, as it will mess with the

**Table 5. Power for standardized effect sizes $\delta$ = 1.0 (top) and $\delta$ = 1.4 (bottom) for combinations of randomization procedures and group sequential design for the z-test.**

| Randomization Procedure | Lan-DeMets O'Brien-Fleming type boundaries | | | Inverse Normal Combination Test O'Brien-Fleming type boundaries | | |
|---|---|---|---|---|---|---|
| Futility | None | Binding | Non-binding | None | Binding | Non-binding |
| Power for standardized effect size of $\delta$ = 1.0 | | | | | | |
| **Complete Randomization** | 0.6629 | 0.6478 | 0.6381 | 0.6227 | 0.6123 | 0.6043 |
| **Permuted Block Randomization** (4) | 0.6819 | 0.6711 | 0.6635 | 0.6819 | 0.6711 | 0.6635 |
| **Big Stick Design** (3) | 0.6800 | 0.6678 | 0.6594 | 0.6577 | 0.6476 | 0.6398 |
| **Random Allocation Rule** | 0.6823 | 0.6668 | 0.6572 | 0.6396 | 0.6291 | 0.6212 |
| **Efron's Biased Coin** (2/3) | 0.6790 | 0.6664 | 0.6579 | 0.6517 | 0.6419 | 0.6340 |
| **Chen's Design** (2/3, 3) | 0.6807 | 0.6691 | 0.6610 | 0.6643 | 0.6542 | 0.6464 |
| Power for standardized effect size of $\delta$ = 1.4 | | | | | | |
| **Complete Randomization** | 0.9145 | 0.8988 | 0.8947 | 0.8870 | 0.8753 | 0.8713 |
| **Permuted Block Randomization** (4) | 0.9264 | 0.9165 | 0.9136 | 0.9264 | 0.9165 | 0.9136 |
| **Big Stick Design** (3) | 0.9254 | 0.9139 | 0.9106 | 0.9121 | 0.9021 | 0.8987 |
| **Random Allocation Rule** | 0.9266 | 0.9114 | 0.9077 | 0.8988 | 0.8877 | 0.8840 |
| **Efron's Biased Coin** (2/3) | 0.9248 | 0.9128 | 0.9094 | 0.9074 | 0.8976 | 0.8941 |
| **Chen's Design** (2/3, 3) | 0.9258 | 0.9149 | 0.9118 | 0.9161 | 0.9064 | 0.9031 |

**Table Description:** Based on 1000 randomization sequences generated from each randomization procedure to calculate the (mean) power. The Table compares scenarios with no futility, binding and non-binding futility boundaries, where we stopped in case of a negative trend, i.e. for stopping boundaries $a_1 = a_2 = 0$ (see S1 Appendix for notation) for a maximum sample size of $n$ = 24 and $K$ = 3 stages of the same size, i.e. two interim analyses and a final analysis. Both Lan-DeMets and inverse normal combination test use O'Brien-Fleming type boundaries for efficacy. The maximum standard error is given by 0.0015 for the inverse normal combination test with non-binding futility. For a description of the randomization procedures see Table 1.

1 : 1 allocation for each stage for PBR. For illustration, for INCT with $K$ = 4 equidistantly distributed stages and a standardized effect size of $\delta$ = 1.0, the power is 50.51% for a maximum sample size of $n$ = 16. Lets say we add an additional patient. However, this patient is added to the first stage, so that we have 5 patients in the first, 4 patients in the second, third and fourth stage. This leads to a power of 49.08% even though we have one more patient for a maximum sample size of $n$ = 17, but the disruption of the 1 : 1 allocation leads to a power loss overall. Even when adding an additional patient in the second stage (for a total of 18 patients), the power remains lower at 50.39% than for $n$ = 16 (the allocations per stage are: 5 in the first and second, 4 in the third and fourth). This effect was evaluated further in S7 Appendix, where we evaluated what happens to PBR, when additional patients get added equally to the stages compared to when they all get added to the final stage.

Thus, when using INCT with small sample sizes, it is essential to strictly maintain the stage-wise sample sizes for small samples. Neglecting to do so may hinder any potential increase in power, which is counterproductive to the ethical goal of enrolling as few patients as possible in clinical trials.

**Results of the simulation study for the t-test.** Table 6 presents the power for a standardized effect sizes of $\delta$ = 1.0 and $\delta$ = 1.4 across different combinations of GSDs and RPs for the t-test, analogous to Table 4 for the z-test. The results show a power loss for the t-test compared to the z-test. The overall trend in regards to the RPs is similar to the z-test, with LDM designs maintaining comparable power across most RPs except for CR, while the power difference between the RPs is more pronounced for INCT designs. For smaller sample sizes the difference between LDM and INCT is also larger, as the stage-wise calculation of t-statistics results in a higher loss of degrees of freedom compared to using LDM. The extent of the power loss between the RPs again depends on the restrictiveness of the RP.

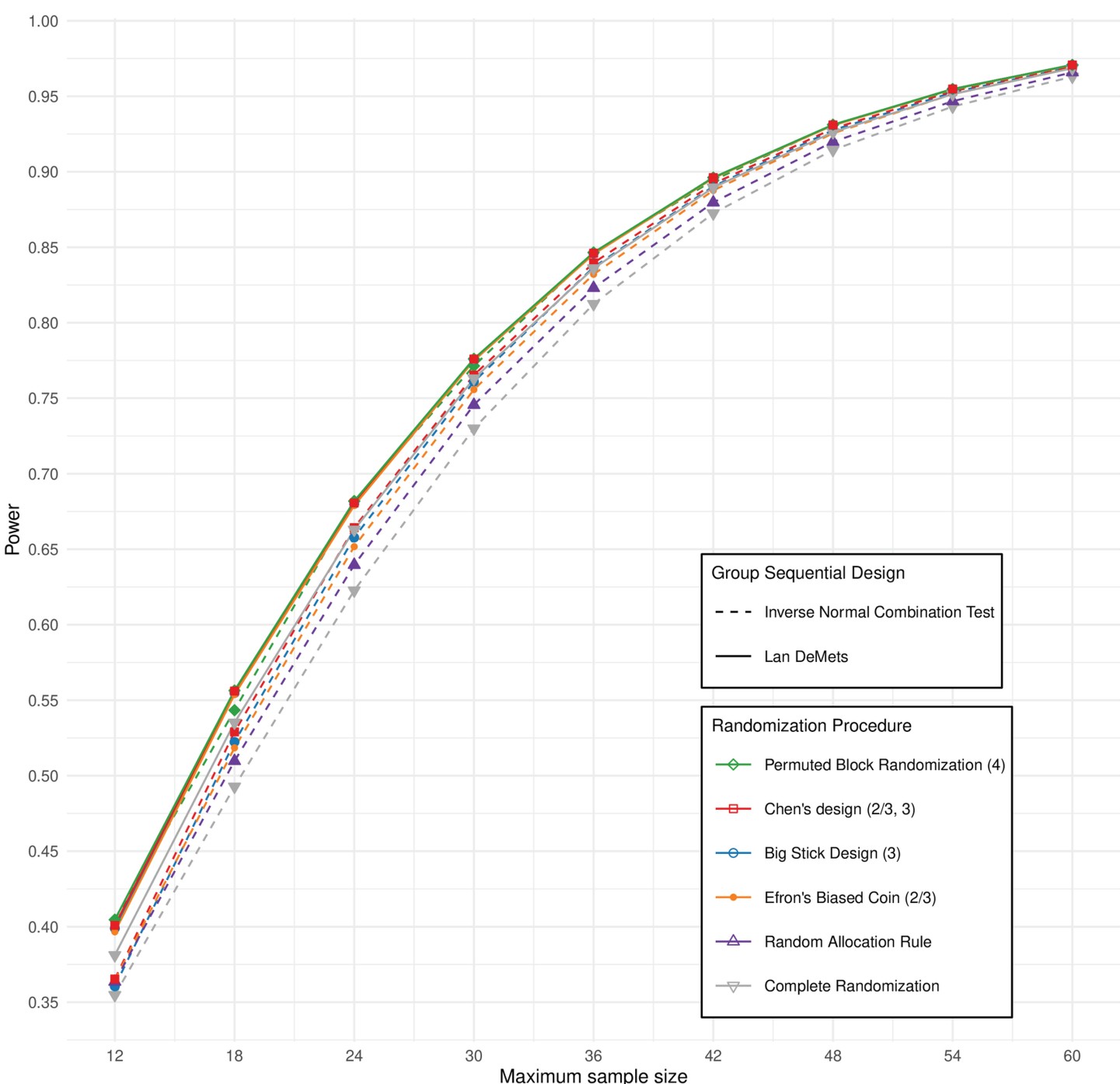

**Fig 5. Power of different randomization procedures for a standardized effect size** $\delta$ = 1.0 **and** $K$ = 3 **stages of equal size in regards to the sample size for the z-test.**
The maximum sample size is depicted on a grid from 12 to 60 increasing in steps of 6. Therefore for every evaluated $n$, the stage-wise sample sizes are all of the same size of $n/3$. The randomization procedures are indicated by different colors. The adjustment method to address repeated significance testing in the group sequential design (GSD) is indicated by solid lines for using a GSD with Lan-DeMets spending functions (LDM) or adaptive combination test using inverse normal function (INCT). Both LDM and INCT use O'Brien-Fleming type boundaries for efficacy. To make the differences visible, the y-axis is shown from 0.35 to 1.

**Table 6. Power for standardized effect sizes of $\delta$ = 1.0 (top) and $\delta$ = 1.4 (bottom) for each combination of randomization procedure and group sequential design for the t-test.**

| Randomization Procedure | Lan-DeMets with O'Brien-Fleming type boundaries | Inverse Normal Combination Test with O'Brien-Fleming type boundaries | Lan-DeMets with Pocock type boundaries | Inverse Normal Combination Test with Pocock type boundaries |
|---|---|---|---|---|
| Power for standardized effect size of $\delta$ = 1.0 | | | | |
| **Complete Randomization** | 0.6249 | 0.5719 | 0.5365 | 0.4789 |
| **Permuted Block Randomization** (4) | 0.6426 | 0.6126 | 0.5536 | 0.5178 |
| **Big Stick Design** (3) | 0.6398 | 0.5894 | 0.5506 | 0.496 |
| **Random Allocation Rule** | 0.6425 | 0.5802 | 0.5540 | 0.4864 |
| **Efron's Biased Coin** (2/3) | 0.6395 | 0.5872 | 0.5503 | 0.4944 |
| **Chen's Design** (2/3, 3) | 0.6410 | 0.5957 | 0.5519 | 0.5021 |
| Power for standardized effect size of $\delta$ = 1.4 | | | | |
| **Complete Randomization** | 0.8898 | 0.8494 | 0.8311 | 0.7762 |
| **Permuted Block Randomization** (4) | 0.9021 | 0.8824 | 0.8462 | 0.8166 |
| **Big Stick Design** (3) | 0.9005 | 0.8644 | 0.8443 | 0.7944 |
| **Random Allocation Rule** | 0.9020 | 0.8565 | 0.8467 | 0.7844 |
| **Efron's Biased Coin** (2/3) | 0.9001 | 0.8628 | 0.8437 | 0.7926 |
| **Chen's Design** (2/3, 3) | 0.9008 | 0.8691 | 0.8447 | 0.8006 |

**Table Description:** Based on 1000 randomization sequences generated from each randomization procedure and 2000 simulated trials for each randomization sequence to calculate the (mean) power. For a maximum sample size of $n$ = 24 and three equally sized stages ($K$ = 3), i.e. two interim analyses and a final analysis. For a description of the randomization procedures see Table 1.

Regarding T1E control, the t-test shows a slight inflation for a maximum sample size of $n$ = 24 and $K$ = 3 stages. Specifically, for the LDM design with OBF type boundaries, mean T1E up to 0.0258 is observed for the RPs and for LDM with Pocock-type boundaries, values as high as 0.0263 are reached. In contrast, the INCT controls the T1E, albeit at the cost of reduced power compared to LDM. Detailed T1E results for these settings are provided in S8 Appendix, along with additional results for a larger maximum sample size of $n$ = 120. In this larger-sample scenario, the power loss associated with the t-test becomes less pronounced due to the increased degrees of freedom, and the stage-wise loss of degrees of freedom for INCT is less consequential. Likewise, the T1E inflation observed with LDM designs at $n$ = 24 is mitigated with the larger sample size.

## Framework for selecting the randomization procedure in small sample group sequential designs

Based on our simulation studies, we summarize key considerations we found during our simulations:

1. When using test statistics based on cumulative data with a pre-defined spending function one has to account for the observed allocation ratio and the potential stage-wise imbalances when calculating the actual critical boundaries to ensure the T1E is controlled even for the z-test (assuming known variances).

2. Avoid RPs that can result in arbitrarily large imbalances within stages, such as CR, EBC and RAR.

3. If the correct spending function incorporating the observed allocation ratio is used, then the power of LDM designs remains robust when using RPs that do not maintain an exact 1 : 1 stage-wise allocation ratio, but achieve a sufficiently close balance, e.g., there is a loss of power for complete randomization (especially for moderate effect sizes).

4. For LDM T1E control becomes tricky. Even when using the quantile substitution method for the t-test the T1E may be inflated, particularly for very small sample sizes, see S8 Appendix.

5. Using the INCT strictly controls the T1E both for z-test and t-test. However, there is a trade-off between strict T1E control and power.

6. When using INCT, RPs that closely maintain the stage-wise 1 : 1 allocation ratio should be preferred to maintain power. For example, when using PBR, select a block size that is a divisor of the stage-wise sample size to meet the targeted allocation ratio and maintain power.

7. For the t-test the INCT has a loss in power compared to LDM. This is more pronounced for smaller sample sizes as in the t-test more degrees of freedom have to be spent for INCT to calculate stage-wise variances and test statistics.

8. For larger stage-wise sample sizes, differences in power, both in regards to the RP and t-test vs. z-test become less relevant (see S6 Appendix, S8 Appendix). However, the INCT would allow for additional features, e.g., sample size reassessment if needed.

In light of these findings, it becomes clear that the choice of RP in combination with the GSD is important in small sample settings. To guide these decisions, we propose a framework that synthesizes the insights from our analysis, including considerations on allocation imbalances, power differences across RPs, and the conditions under which PBR must be aligned with stage-wise sample sizes for optimal power. Table 7 shows this framework, which is relevant for small stage-wise sample sizes, say a (minimum) stage-wise sample size that is smaller than 20.

## Discussion

Our study highlights that the choice of RP in small sample GSDs affects the trials outcome. This effect is noticeable even under ideal conditions, such as when variance is known. Table 7 provides a framework indicating suitable RPs to pair with GSDs for trials with small stage-wise sample sizes. The framework is applied to equally distributed stages, as evaluated in our simulation, but it can be straightforward extended to accommodate varying interim analysis timing.

Our simulation shows, that complete randomization, though conceptually simple, leads to a power loss for small sample GSDs . To mitigate this issue, restricted randomization methods are commonly employed to achieve balanced treatment groups sizes with respect to final analysis (final balance) as well as stage analysis (stage-wise balance). If no further restriction, like stratification of the RP by stage is implemented, the only procedure which achieves stage-wise balance is PBR by using according block sizes. However, PBR in unblinded or single-blinded trials, which are common in fields such as rare diseases [23], carries a high risk of predictability [24]. For example, in a trial with a maximum sample size of $n = 24$ , an investigator could correctly predict at least 1/4 of the allocations with certainty, if he correctly assumes that PBR(4) was used, the most common used RP and block size in our literature review on GSDs,  or simply follows the "convergence strategy" for guessing the next treatment allocation

**Table 7. Framework for selecting the randomization procedure in small sample group sequential designs for an intended** $1:1$ **allocation ratio with equal stage-wise sample sizes.**

| Randomization Procedure | Lan-DeMets | Inverse Normal | Standard Boundaries* |
|---|---|---|---|
| **Complete Randomization** | Low power; risk of allocations to single group | Low power; risk of allocations to single group | Type I error rate (T1E) not controlled |
| **Random Allocation Rule** | Risk of allocations to single group | Low power; risk of allocations to single group | T1E not controlled |
| **Big Stick Design ($m$)** | For $m \le \frac{n}{K}$ | For $m \le \frac{2n}{K}$; power reduction relative to $1:1$ allocation per stage | T1E not controlled |
| **Permuted Block Randomization ($l$)** | | $l$ should divide $\frac{n}{K}$ for optimal power | $l$ must divide $\frac{n}{K}$ to control T1E |
| **Efron's Biased Coin ($p$)** | Risk of allocations to single group | Moderate power; risk of allocations to single group | T1E not controlled |
| **Chen's Design ($p, m$)** | For $m \le \frac{n}{K}$ | For $m \le \frac{2n}{K}$; power reduction relative to $1:1$ allocation per stage | T1E not controlled |

**Table Description:** The maximum sample size is $n$ and $K$ is the number of pre-planned stages. Therefore the stage-wise sample size is given by $n/K$. The framework is intended for cases with small stage-wise sample sizes (say $n/K \le 20$). For the inverse normal combination test, weights proportional to the pre-planned stage-wise sample sizes were used. The description of randomization procedures can be found in Table 1.
* With standard boundaries we refer to the case where the boundaries are calculated assuming an $1:1$ allocation ratio.

[26]. This predictability introduces a risk of allocation bias, which could undermine the trials validity.

RAR, a special form of block randomization with the block size being the total sample size, can keep the final balance, but does not maintain the stage-wise balance. This procedure was shown in our simulations to perform sub-optimally in regards to power for INCT in small sample GSDs, as it does not keep the stage-wise balance. Other RPs, like BSD and CHEN keep a maximum tolerated imbalance, where the imbalance at each time of the allocation process never exceeds a certain threshold. This is particularly useful in small sample clinical trials, as higher deviations from the stage-wise $1:1$ allocation ratio are prohibited.

The last category of RPs are those that can become arbitrarily imbalanced, which is very problematic in small sample group sequential clinical trials, namely CR and EBC.

It should be noted, that we evaluated some special candidates of RPs, showing the property of stage-wise balance, final balance only, keeping the maximal tolerated imbalance only, as well as having no restrictions, to mirror the picture of these properties. The randomizeR package can generate sequences for other RPs [40], and these can be used together with our code [45] to evaluate further RPs.

The literature reflects an ongoing debate regarding the appropriate choice of RPs and for which scenarios PBR is appropriate [20]. The EMA ICH E9 guideline states that the investigator should be blinded to the block size [8], however, this approach presents challenges in GSDs, where block sizes are usually divisors of the stage-wise sample sizes to accommodate the $1:1$ allocation during (interim) analyses.

Some RPs can be modified, so that their restrictions apply for each stage to enforce a close $1:1$ balance not overall, but stage-wise, as in stratified randomization. However the modifications have to be implemented in such a way that the overall balance is not negatively affected. For example if the same maximum tolerated imbalance would be applied in a stage-wise manner, the actual tolerated one would accumulate. The disadvantage is, that this leads to even more restrictive RPs. Enforcing a $1:1$ allocation at the end of each stage is mathematically

equivalent to employing PBR with a block size equal to the stage size as an additional restriction, which increases the predictability and allocation bias risk. For the INCT design, restarting the RP at the beginning of each stage may be beneficial, as tests are conducted independently at each stage rather than cumulatively across all previously allocated patients, as is the case in the LDM design.

In contexts where limited information is available, such as in rare diseases, INCT designs may offer advantages through their ability to reassess sample sizes during the trial. However, one should carefully evaluate if adapting the sample size is really necessary. Otherwise, LDM should be preferred in small samples due to the power loss associated with INCT compared to LDM. Furthermore, the decision which design to use and what RP to choose is accompanied by the extend of power loss acceptable, which depends on the trial settings and the cost of additional patients from a cost, availability and time perspective.

While simulation studies generally assume a 1 : 1 allocation at each stage for comparing various GSDs, this assumption may not hold in practice, particularly in small trials where deviations from balanced allocation have a larger impact.

Given the substantial risk of allocation bias, a more systematic approach to evaluating its effects in GSDs is needed. Extending frameworks such as the ERDO framework to incorporate allocation bias considerations will be a valuable step forward [24]. Moreover, since these problems are already evident in "simpler" adaptive trials, they are likely to be more pronounced in more complex designs, such as platform trials [46,47]. This can be even more relevant in designs where the allocation ratio may change depending on the number of active treatments running in parallel [48] or when using more complex methods such as response adaptive randomization [49–51]. Therefore, further research into the performance of different RPs in these complex designs is warranted.

Conducting the literature review for this study was challenging due to ambiguous terminology and inconsistent reporting of GSDs. For example, the term "O'Brien & Fleming boundaries" could refer to the traditional OBF design, an INCT test with OBF stopping boundaries, or even a LDM method inaccurately reported. Statements such as "O'Brien & Fleming function is used", accompanied by a citation to the original OBF paper, fail to specify which method was applied. Such vague descriptions make it difficult to identify the exact GSD being used. To improve reproducibility and transparency, we recommend that future reports on GSDs clearly specify the type of GSD, the exact stopping boundaries (both efficacy and futility), and the number of participants allocated to each group at each stage. This issue is compounded by poor reporting practices for RPs. Often, details of RPs and their parameters are either not reported or buried within the trial's statistical analysis plans, making them difficult to locate. Adherence to the consolidated standards of reporting trials (CONSORT) guidelines, which call for transparent reporting of randomization and adaptive design features, is crucial to address these gaps [22,52].

## Conclusion

Accurate and detailed reporting of GSDs, including the specific stopping boundaries and RPs with their parameters, is crucial for ensuring reproducibility and transparency in clinical trials [22,53]. However, these elements remain under-reported, particularly in trials that employ GSDs. This gap in reporting is especially concerning in small-sample trials, such as those in rare diseases, where the choice of RP has a significant impact on trial outcomes, especially when sample size reassessment is involved.

We developed clear recommendations which RP should be used in connection with the GSD. The LDM approach might be favored to the INCT in small-sample settings, as it has

higher power when the variance is unknown, i.e., for the t-test and is more resistant to deviations from expected allocations at each stage, which might also occur from overrunning. As an alternative to PBR, RPs like CHEN or BSD with a low maximum tolerated imbalance, should be considered to reduce the predictability of the RP, thereby reducing the risk of allocation bias. Nonetheless, INCT remains advantageous when sample size reassessment is necessary.

## Supporting information

**S1 Apppendix. Group sequential designs using standard Pocock, O'Brien-Fleming and Lan-DeMets adjustments.** In this technical appendix we describe how we calculated the type I error rate and the power conditioned on a randomization sequence for boundaries from the standard Pocock, O'Brien-Fleming and Lan-DeMets designs for a z-test.
(PDF)

**S2 Appendix. Group sequential design with inverse normal combination test.** In this technical appendix we describe how we calculated the type I error rate and the power conditioned on a randomization sequence for the inverse normal combination test for a z-test.
(PDF)

**S3 Appendix. Investigated values and operating characteristics of the simulation for the z-test.** This appendix provides a comprehensive overview of all simulation settings explored for the z-test, including the operating characteristics evaluated. Additionally, we detail the number of randomization sequences generated that resulted in all allocations being assigned to a single group during a stage.
(PDF)

**S4 Appendix. Type I error for different combinations of group sequential designs and randomization procedures using the z-test.** This appendix shows the violin plots of the type I error for Lan-DeMets and inverse normal combination tests.
(PDF)

**S5 Appendix. Power for different combinations of group sequential designs and randomization procedures using the z-test.** This appendix extends the power analysis presented in the main manuscript by showing the power for additional group sequential designs.
(PDF)

**S6 Appendix. Different maximum sample sizes and number of stages for group sequential designs using the z-test.** This appendix extends the power analysis presented in the main manuscript by exploring different maximum sample sizes.
(PDF)

**S7 Appendix. Permuted block randomization for different maximum sample sizes for group sequential designs using the z-test.** This appendix shows the power of permuted block randomization for different maximum sample sizes, when the stage-wise sample sizes are not necessarily divisible by the block length(s).
(PDF)

**S8 Appendix. t-test for group sequential designs.** This appendix shows the methodology for the t-test in group sequential designs, describes how the simulation was conducted for the t-test, presents the type I error rate for a maximum sample size of $n = 24$ and $K = 3$ stages and

presents additional results for a maximum sample size of $n = 120$ and $K = 3$ stages.
(PDF)

**S9 Appendix. Validation for z-test results.** This appendix presents a comparison between z-test results obtained via the multivariate normal method and those derived from trial simulations.
(PDF)

**S10 Appendix. Mean absolute imbalance in number of patients for each treatment group by randomization procedure.** This appendix presents the mean absolute imbalance of group sizes for each interim and final test across different randomization procedures for a maximum sample size of $n = 24$ and $K = 3$ equidistant interim analyses.
(PDF)

**S11 Appendix. Excel file corresponding to literature review.** This Excel file shows a list of all studies and their corresponding settings included in our literature review.
(XSLX)

## Acknowledgement

We thank Chris Jennison for contributing code to calculate group sequential boundary properties, which was used for the initial power calculations in this work and gratefully acknowledge the computing time provided to at the NHR Center NHR4CES at RWTH Aachen University (project number p0024408).

## Author contributions

**Conceptualization:** Daniel Bodden, Ralf-Dieter Hilgers, Franz König.

**Data curation:** Daniel Bodden.

**Formal analysis:** Daniel Bodden.

**Funding acquisition:** Ralf-Dieter Hilgers.

**Investigation:** Daniel Bodden.

**Methodology:** Ralf-Dieter Hilgers, Franz König.

**Software:** Daniel Bodden.

**Supervision:** Ralf-Dieter Hilgers, Franz König.

**Visualization:** Daniel Bodden.

**Writing – original draft:** Daniel Bodden.

**Writing – review & editing:** Daniel Bodden, Ralf-Dieter Hilgers, Franz König.

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
