## [Decision Letter · Decision Letter 0]

18 Feb 2025

PONE-D-24-53742Randomization in clinical trials with small sample sizes using group sequential designsPLOS ONE

Dear Dr. Bodden,

Thank you for submitting your manuscript to PLOS ONE. After careful consideration, we feel that it has merit but does not fully meet PLOS ONE’s publication criteria as it currently stands. Therefore, we invite you to submit a revised version of the manuscript that addresses the points raised during the review process.

We look forward to receiving your revised manuscript.

Kind regards,

Dhermendra Tiwari

Academic Editor

PLOS ONE

2. Thank you for stating the following financial disclosure:  [RDH received funding from European Rare Disease Research Coordination and Support Action consortium (ERICA) funded by the European Union’s Horizon 2020 research and innovation programme under Grant Agreement. no. 964908. Further the work of FK and RDH was supported by the RealiseD project funded under Innovative Health Initiative under Grant Agreement. no. 101165912. The content of the paper reflect the personal view of the authors.].  Please state what role the funders took in the study.  If the funders had no role, please state: "The funders had no role in study design, data collection and analysis, decision to publish, or preparation of the manuscript." If this statement is not correct you must amend it as needed. Please include this amended Role of Funder statement in your cover letter; we will change the online submission form on your behalf.

3. Thank you for uploading your study's underlying data set. Unfortunately, the repository you have noted in your Data Availability statement does not qualify as an acceptable data repository according to PLOS's standards.

At this time, please upload the minimal data set necessary to replicate your study's findings to a stable, public repository (such as figshare or Dryad) and provide us with the relevant URLs, DOIs, or accession numbers that may be used to access these data. For a list of recommended repositories and additional information on PLOS standards for data deposition, please see https://journals.plos.org/plosone/s/recommended-repositories

Additional Editor Comments (if provided):

Reviewers' comments:

Reviewer's Responses to Questions

**Comments to the Author**

1. Is the manuscript technically sound, and do the data support the conclusions?

Reviewer #1: Yes

Reviewer #2: Yes

Reviewer #3: Yes

Reviewer #4: Partly

2. Has the statistical analysis been performed appropriately and rigorously? 

Reviewer #1: Yes

Reviewer #2: Yes

Reviewer #3: Yes

Reviewer #4: Yes

3. Have the authors made all data underlying the findings in their manuscript fully available?

Reviewer #1: Yes

Reviewer #2: Yes

Reviewer #3: Yes

Reviewer #4: Yes

4. Is the manuscript presented in an intelligible fashion and written in standard English?

Reviewer #1: Yes

Reviewer #2: Yes

Reviewer #3: Yes

Reviewer #4: Yes

5. Review Comments to the Author

Reviewer #1: Reviewer: Michael Grayling, J&J

Summary

Thanks for the opportunity to review this paper. There is a lot to like here: the authors have spotted something that has (curiously) not been discussed in the group-sequential design literature. They have then taken comprehensive approach to demonstrating this is a real issue through a literature review, and helped propose solution(s) by evaluating several possible randomisation procedures that could be utilised in practice. The manuscript is also very well written and the accompanying code is well-commented, which will heavily assist any future users. I only have one suggestion for something to add to the Discussion.

Comments

1. You touch a little on assuming known variance in the Discussion, but I think you need to go slightly further on this point to help any readers understand what they’d need to do in practice in this setting. The sample sizes considered are evidently very small. In a real study, no one assumes a z-test; when using a t-test, methodology would almost certainly be needed to adjust the boundaries computed under a z-test assumption to prevent type I error inflation even in the case of perfect balance. Something like https://pubmed.ncbi.nlm.nih.gov/17434814/ or the quantile substitution approach discussed in https://pubmed.ncbi.nlm.nih.gov/18642403/ would likely be needed. So. arguably the paper is commenting on the relative inflation levels with different randomisation procedures, rather than the absolute inflation levels of each procedure (which I think could only be known with a simulation study, rather than any use of multivariate normality assumptions).

2. Some typos I spotted:

- Abstract: “in in”

- Left quotation marks need to be inputted as `` in latex.

- Line 111: K=3 to $K=3$

Reviewer #2: This is an interesting study. The authors investigated how randomization procedures have an impact on the results of small clinical trials using group sequential designs. Below are a few comments listed.

Small clinical trials are generally not required to have an interim analysis as the small clinical trial is not confirmatory study. It’s not very clear to me if traditional group sequential design or adaptive trial design was being studied in the paper. The traditional group sequential designs are commonly used in large clinical trials.

The authors provided the results obtained from the literature review (Table 3). The study is focused on the small clinical trials. I wonder if the sample size of each study can be added into the table to show the information.

P. 3, line 73. “they are particularly relevant for RD clinical trials”. What does the “RD” mean?

Reviewer #3: This paper evaluate how different randomization procedures affect the type I errorr and power in small sample group sequential clinical trials. The topic is interesting and relevant and the paper is well written.

I have only a few minor comments:

- Abstract: In the result section: ‘deficiencies in in the implementation’: delete 1 ‘in’ ; Additionally, the title of the article and the objective on p2. line 45 clearly focus on the small sample setting, which is barely mentioned in the abstract – I recommend to make it also clear in the abstract that the focus is small sample trials.

- p. 1 line 32-34: ‘With respect to application of GSD’s trials reported from 2001 to 2014 the LDM design was used in 34% with an OBF error spending function and 27% used the traditional OBF design’. These percentages do not add up to 100%, which makes me wonder which error spending function is used in the remaining 39% (which is the majority of cases!).

- p. 3 line 72: ‘In comparison to Stevely et al. [11], we also included randomized Phase II clinical trials utilizing GSDs, as they are particularly relevant for RD clinical trials’ – The abbreviation RD has not been introduced. Does it stand for randomized, or rare disease,…? Whatever it means, I feel the last part of that sentences is odd. If RD means randomized, then I fail to see that, if GSD is relevant for randomized clinical trials, you include only phase II trials. Additionally, do the authors mean with ‘in comparison’ that they follow the set-up of Stevely or do they mean ‘in contrast’?

- p. 3 line 76: The section that starts at this line seems out of place. Is this part of the simulation? It does not seem part of the literature review, since single center, normal endpoint trials were not part of the search criteria. Similar comment for the section starting at line 91. Perhaps they can be part of the introduction or part of the simulation section, but now these sections are disconnected and out of place.

- Table 3: GSD section, the categories sum to 67, but n=71. On p. 8 line 190-191 it is stated that ‘These percentages do not sum up to 100% as some studies utilized multiple group sequential boundaries, for instance, for different endpoints.’. If multiple methods have been used, I expect a sum larger than 100%, not below 100%. Additionally, the ‘Information on block length’ = 14, but the below categories sum only to 13.

- p. 8 line 190: based on Table 3 should it not read: ‘and 3% for the Pocock design’, rather than 2%?

- p.8 line 206-207: Please revise the following sentence, which is grammatically not correct: ‘The largest inflation in mean T1E is observed for RAR, as balanced allocation at interim analysis does not always result, although RAR leads to an overall 1 : 1 allocation by the end of the trial. ‘

- p. 9 line 234: it is not clear what is meant with 2a in ‘resulting in 2a consecutive allocations to the same group’.

- p. 12 line 334: suiTable should be suitable

Reviewer #4: The authors evaluated the effect of different randomization procedures on the type-I-error probability and power of a group sequential two arm parallel group design with an intended allocation ratio of 1:1 and a continuous normally distributed endpoint. The evaluation is based on an extensive simulation study combining different approaches to alpha-spending, determination of stopping boundaries, and randomization procedures for different sample sizes, using a one-tailed z-test at a one-sided level of 0.025 with a known variance of 1.

Special attention is paid to small sample sizes. This raises the question of whether the z-test, which is often recommended in the literature only for sample sizes greater than 30, is appropriate. To justify this, it would have to be argued - at least by way of example - that the t-test and the z-test give comparable results even with small sample sizes, or that the evaluation is based entirely on the t-test. The R-package rpact could be helpful here.

Another important aspect of the simulation analyses in terms of power assessment is the effect size to be detected, i.e. the number of patients required to achieve a target power or, conversely, to achieve a specific power for a given sample size. This raises the question of a justification for the choice of the effect sizes of 1.0 and 1.4 or vice versa, what effect can be detected with a sample size of 24 patients.

In the discussion, the authors argued that there was a substantial power loss – due to the deviation of the intended 1:1 allocation ratio at interim analyses - but without quantifying it. It may be helpful to discuss what level of power loss is still acceptable and where it becomes so substantial that the validity of the study is questionable. Since both the power reduction and the inflated type-I-error probability are related to the deviation from the 1:1 allocation, information on the stage-wise deviation from 1:1 would be valuable.

The authors note that insufficiencies in the implementation of randomization can lead to an increase in the type I error rate. Furthermore, some combinations of sequential group designs and randomization procedures lead to a loss of power. To guide the selection of a suitable combinations of a sequential group design and a randomization procedure, the authors provide an overview of the combination under consideration, together with a traffic light rating. This framework seems to be the key message of the manuscript and should be pronounced more strongly. Further explanations and practical recommendations (in addition to the bullet list on page 12) would be useful. E.g. information on type-I-error probability and power for each combination (even if it can be seen in the appendix, it would be nice for a “framework” to have all the information bundled in one place). Another point would be, for example that for the Random Allocation Rule and Efron’s Biased Coin Design the “risk of allocations to single group” is only present if the randomization procedure is applied over all stages. This risk cannot occur if the procedure is implemented stages-wise.

Randomization procedures differ in their approach to treatment allocation. While some procedures do not aim for a final balance, others aim for a balance at the end of the allocation process or at defined interim steps. It would be helpful to classify and differentiate between the procedures under consideration in terms of balance, also to be able to assess further procedures. It goes without saying that there are other procedures.

Minor comments

“Simulation results demonstrate that deficiencies in in the implementation of randomization can inflate type I error rates.” What exactly is meant by “deficiencies”? It would be good to see at least one example of this shortcoming and possible measures to prevent it.

The description in Table 1 is (mainly) based on the case of two treatments. This should be made explicit or the description should be at a general level.

The term “sequence” should be added to Table 1

“In the case of randomized PBR” should be explained. Is permuted block randomization with varying block sizes meant here?

“This is important because an imbalance in one stage can reverse in a later stage, resulting in 2a consecutive allocations to the same group.” What is meant by “2a”?

Page 12, line 333: “Table 6 provides a framework indicating suiTable RPs to pair with GSDs for trials with small, stage-wise sample sizes.” – typo T in suiTable

6. PLOS authors have the option to publish the peer review history of their article (what does this mean?). If published, this will include your full peer review and any attached files.

Reviewer #1: **Yes: **Michael Grayling

Reviewer #2: No

Reviewer #3: No

Reviewer #4: No

---

## [Author Response · Author response to Decision Letter 1]

7 Apr 2025

Please find the point-to-point answers to all reviewer comments in the "Response to Reviewers" pdf-file attached.

---

## [Editor Report · Decision Letter 1]

12 May 2025

Randomization in clinical trials with small sample sizes using group sequential designs

PONE-D-24-53742R1

Dear Dr. Daniel Bodden,

We’re pleased to inform you that your manuscript has been judged scientifically suitable for publication and will be formally accepted for publication once it meets all outstanding technical requirements.

Kind regards,

Dhermendra Tiwari

Academic Editor

PLOS ONE
---

## [Editor Report · Acceptance letter]

PONE-D-24-53742R1

PLOS ONE

Dear Dr. Bodden,

I'm pleased to inform you that your manuscript has been deemed suitable for publication in PLOS ONE. Congratulations! Your manuscript is now being handed over to our production team.

Kind regards,

on behalf of

Dr. Dhermendra Tiwari

Academic Editor

PLOS ONE